# Stacking in Layered Covalent Organic Frameworks: A Computational Approach and PXRD Reference Guide

**DOI:** 10.3390/ijms26189222

**Published:** 2025-09-21

**Authors:** Robbin Steentjes, Egbert Zojer

**Affiliations:** Institute of Solid State Physics, NAWI Graz, Graz University of Technology, Petersgasse 16, 8010 Graz, Austria

**Keywords:** covalent organic frameworks, LCOF, stacking, disorder, structural model, ab initio, density functional theory

## Abstract

The stacking arrangement of layered covalent organic frameworks (LCOFs) critically influences their structure and function. We present a fully ab initio-based workflow to characterize stacking disorder in COF-1, combining simulated powder X-ray diffraction (PXRD) with stacking energy landscape analysis. By comparing PXRD patterns of idealized eclipsed, inclined, serrated, and staggered stacking with experiment, we rule out periodic high-symmetry motifs. A comprehensive “PXRD reference guide” links specific diffraction features to slip directions and magnitudes, providing a blueprint for the interpretation of experimental data of slipped structures. Quantum-mechanical potential energy surfaces reveal multiple symmetry-equivalent minima separated by small barriers. This makes diverse slip configurations thermally accessible and large-scale stacking disorder inevitable. Nevertheless, as staggered configurations are found to be energetically disfavored, open pore channels prevail despite the disorder. From the energy landscapes, we construct static disordered models using Boltzmann-weighted probabilities, where also the question is addressed, which energies should be used for actually calculating the Boltzmann weights. Simulated PXRD patterns from these models excellently reproduce experimental peak positions, shapes, and stacking distances, suggesting the dominance of disordered stacking not only in COF-1.

## 1. Introduction

Since their discovery in 2005, two-dimensional covalent organic frameworks (2D COFs) or, more precisely, layered COFs (LCOFs) have gained academic interest and practical impact [1]. Due to their low density, high internal surface area, and large pores, they find applications in gas separation, absorption, and storage [1,2,3,4,5,6,7,8]. More recently, their π-conjugation has led to proposed applications in (opto-)electronics [9,10].

LCOFs consist of two-dimensional covalently bound networks (illustrated for the prototypical system COF-1 in Figure 1), which are stacked in the third dimension due to non-covalent forces, such that 3D crystals are formed [11,12]. The stacking arrangement influences the internal surface area, pore shape and volume, as well as the electronic structure of the material. This can greatly affect the material’s performance in application [13,14,15]. Due to an interplay of forces, the stacking is far from straightforward. Slight differences in the structure of the 2D network can have a major impact on the preferred stacking motifs within a crystal. To understand the characteristics of LCOFs, and to enable reliable theoretical property predictions, it would, therefore, be helpful to have a consistent method to construct models that accurately represent LCOFs.

Following their discovery, LCOFs were expected to assume idealized conformations [1]. The simplest, and most symmetric conformation is ‘eclipsed’ stacking (Figure 2a). Here, the 2D layers are placed directly on top of each other, such that, when viewed from the top, the layers eclipse each other. The main geometric parameter for this stacking is the stacking distance *d_z_*, which describes the distance between consecutive layers.

A possibility to move away from the highly symmetric eclipsed structure is to introduce an in-plane slip ***s*** = (*s_x_*, *s_y_*) (Figure 1). Such a slip indicates that a layer’s origin is displaced by a vector *(s_x_, s_y_, d_z_)* relative to the underlying layer. If each layer has the same nonzero slip, the structure is called ‘inclined’ (in the literature this is also referred to as unidirectional slip stacking; see Figure 2b). An eclipsed stacking mode can be seen as a special case of an inclined mode, where the slip is zero. In passing we note that inclined stacking modes are the only possible outcome of theoretical geometry optimizations, when the input cell contains only one layer of the material.

Another special case of an inclined structure is the ‘staggered’ structure. Here, the node of one layer is on top of the center of the pore of another layer. For systems with hexagonal symmetry, there are two ways to fulfill this criterion, which correspond to ABABAB- and ABCABC-type stacking. This is illustrated in Figure 2d,e, and Appendix A. As this succession is somewhat reminiscent of the stacking in densely packed metals, we will in the following refer to these stacking motifs as HCP-like and FCC-like, respectively [16] (pp. 15–16). For the HCP-like mode, the unit cell contains two layers and the slip between the first and the second layer of the unit cell amounts to ***s***_12_ = ⅔***a*** + ⅓***b*** (Figure 2d), with ***a*** and ***b*** representing the in-plane unit-cell vectors (Figure 1). The slip between the second layer and the first layer of the next unit cell amounts to ***s***_21_ = −***s***_12_. In this way, every other layer is eclipsed. For FCC-like staggered structures, the unit cell can be described as containing three layers, each having the same slip ***s***_12_ = ***s***_23_ = ***s***_31_ = ⅔***a*** + ⅓***b***. For illustration purposes such a three-layer periodic honeycomb lattice is shown in Figure 2. It should be noted in passing that due to the equivalent inter-layer slips in the FCC-like staggered structure, an equivalent (and from a crystallographic point of view more appropriate) description of this structure would be an inclined unit cell comprising a single layer with the slip set to ***s*** = 2/3***a*** + 1/3***b*** and a stacking distance *d_z_*. A more detailed illustration of the difference between the two staggered stacking modes is provided in Appendix A.

A ‘serrated’ structure (also referred to as an alternatingly slip-stacked structure) is analogous to an HCP-like staggered structure, the only difference being an arbitrary direction and magnitude of the slip between the two layers in the unit cell (Figure 2c). Still, every other layer is eclipsed, as the two slips between consecutive layers alternate between ***s*** and −***s***. In that sense, the serrated stacking mode differs from the inclined mode by the alternating sign of the slip vector.

There is a priori no necessity for a crystal to assume one single stacking motif. Usually, rather weak van der Waals forces dominate the interactions between layers. As these forces are comparably short-ranged, the influence of the specific arrangement of two layers will often only have a rather minor impact on the arrangement of the next layers. Moreover, there will typically exist multiple slips corresponding to local minimum structures, which, in the case of the hexagonally symmetric COF-1, all have a six-fold rotational symmetry. For systems comprising building blocks with reduced symmetry, also the number of equivalent minima will decrease correspondingly. Considering entropy as an additional driving force, the thermal motion of consecutive layers especially during crystal growth and also within the final crystal, will most likely result in a disordered, non-periodic series of slip-vectors ***s****_i_*. Thus, we propose that in any LCOF crystal that contains more than one stable energetic minimum, a certain stacking disorder is inevitable.

This poses challenges both for the simulation as well as for the experimental characterization of LCOFs. In classical geometry optimizations, lattice parameters and atomic positions converge to a single (often local) minimum structure. In the case of several local minima, the structure obtained in an optimization is typically determined by the starting configuration. To obtain a more comprehensive picture, we will use a combination of machine learning and selected quantum-mechanical simulations to determine the total energy and stacking distance as a function of the slip. Then, the whole energy landscape including multiple minimum structures can be considered when determining disorder in the layer stacking.

To experimentally analyze LCOF structures, one would ideally apply single-crystal diffraction experiments to sufficiently large COF crystals. These are, however hard to obtain and, thus, only a few years ago the first such study has been reported [17]. As an alternative approach, when studying thin films rather than large crystals, rotating grazing incidence diffraction performed at synchrotrons can be a viable tool for generating a picture of large fractions of reciprocal space of porous materials [18] and the technique could also be applied to COF thin films. Unfortunately, however, COFs are usually synthesized as powders comprising small crystallites [19]. Moreover, due to the aforementioned stacking disorder, single crystals or thin films with full 3D periodicity are an unlikely scenario for many layered COFs. Thus, powder X-ray diffraction (PXRD) experiments are still the most commonly applied technique and also in the current manuscript we will rely on comparing our simulations to that type of experimental data [20,21]. In passing we note that ultra-high-resolution synchrotron radiation diffraction could be advantageous to avoid missing and overlapping diffraction signals.

The PXRD experiments yield a set of peaks at certain scattering angles (2θ), with specific intensities, widths, and shapes. In perfect crystals, peaks are caused by clearly defined reflection planes, denoted by their Miller indices [22]. When disorder is present, the resulting diffractogram is effectively an average of scattering on all stacking modes or phases present in the sample. This is not quite the same as scattering on an average structure, a misunderstanding that can lead to a misinterpretation of data [23,24]. In the following, we will illustrate that PXRD data still provide valuable insights into disorder.

Previous efforts to model stacking in layered COFs have employed a variety of approaches, reflecting the complexity of the problem. Early studies often focused on idealized, high-symmetry stacking arrangements such as eclipsed or staggered configurations [1,25], which fail to account for the disorder commonly observed in experimental samples. Soon after, lower symmetry, but still idealized stacking modes were proposed, such as inclined and serrated structures [20]. More recently, the focus has shifted to include forms of stacking disorder. Experimental investigations by Pütz et al. using total scattering and pair distribution function analysis revealed random layer offsets in TTI-COFs, highlighting the risk of an incorrect interpretation of diffraction data in some previous studies insofar as “random slip stacking is easily misinterpreted as apparent eclipsed stacking” [23]. A theoretical investigation by Zhang et al. showed that models including statistical distributions of stacking modes reproduce experimental PXRD patterns far better than idealized crystalline models [24].

These studies underscore the importance of explicitly including stacking disorder in structural models to realistically represent the crystallographic features and functional behavior of LCOFs.

In this work, we acknowledge that disorder is fundamental in determining the structure of LCOFs. Here, we propose a method to accurately capture this disorder, based solely on ab initio calculations. We show that the theoretically determined model, which includes no empirical data or expectation bias, yields excellent agreement with experimentally obtained powder X-ray diffraction (PXRD) data. This is in particular shown for COF-1, as the prototypical honeycomb LCOF system (Figure 1), where a conceptually equivalent situation will prevail also for, e.g., square lattices [24]. Moreover, we give insight into the influence certain stacking modes have on PXRD patterns, by providing a library of PXRD patterns for hypothetical stacking configurations. This will serve as a useful reference for determining stacking motives of hitherto unknown LCOF structures based on experimental PXRD data.

## 2. Results

### 2.1. Inconsistency of Idealized Stacking Modes with Experimental Observations

As a first step, the experimental observations for COF-1 by Côté et al. are compared to PXRD data of the idealized stacking modes described in Figure 2 [1]. Details on the calculations of PXRD spectra, and the generation of the structural models via (constrained) geometry optimization processes, can be found in the methods section. Figure 3 displays the simulated curves for the eclipsed, inclined, serrated, HCP-like staggered, and FCC-like staggered configurations. For the comparison between experimental and simulated peaks, we will use the four-index Miller-Bravais notation for hexagonal systems, where peaks are denoted by *(hkil)*, where the third index is the redundant parameter *i =* −*(h + k)* [26].

In the experiment, the first clearly resolved peak occurs at 2θ = 6.8°. The shape of this peak is highly asymmetric with a pronounced shoulder at larger scattering angles, an aspect that will become relevant later. When ***c*** is perpendicular to the ***ab***-plane, this peak would correspond to scattering from the {10-10} (= (10-10), (01-10), (1-100)) planes all at once. The reason for that is that they are characterized by identical inter-plane distances, as illustrated in Figure 4. Such a situation is encountered for eclipsed, serrated, and staggered structures. For inclined structures, ***c*** is not perpendicular to the ***ab***-plane, changing the scattering angles of the family of {10-10} peaks. As discussed below, in some cases, all peaks from this family shift identically, preserving the overlap. In others, the symmetry is further reduced, and the main peak splits [23]. One can clearly observe this for the inclined structure in Figure 3, for which the slip, resulting from a geometry optimization as described in the Methods section, amounts to ***s*** = (3.7, 1.0) Å. The origin of the shift of {10-10} peaks in inclined structures can be inferred from Figure 4 and is illustrated explicitly in Appendix A. Since for the optimized inclined structure, three distinct peaks in the region around 2θ = 6.8° appear, while in the experimental curve only one strong peak is visible, an inclined configuration is incompatible with the experimental situation and can, thus, be excluded.

In certain cases, the {10-10} peaks can even vanish due to symmetry. This happens, for example, when interstitial layers between reflection planes yield destructive interference of reflected X-rays, which results in the scattering amplitude becoming zero. This is the case for FCC-like staggered stacking, as is illustrated in Appendix A. The Appendix A also contains a mathematical derivation of this observation based on the structure factor for a hexagonal crystal. In contrast, for HCP-like staggered stacking, there would be an additional strong (10-11) reflection at 2θ = 16°. Neither the vanishing of {10-10} peaks nor the appearance of a strong (10-11) reflection are consistent with the experiment. Thus, both staggered stacking conformations can be ruled out for COF-1.

This leaves the eclipsed and serrated structures as possible candidates. These, at first sight, produce diffractograms that fit reasonably well to the experiment [23]. The analysis of the so-called stacking peak around 2θ = 27° (which is directly related to the inter-layer distance *d_z_*) highlights some important differences between these two structures. First, due to the differently defined unit cells, the stacking peak does not have the same Miller indices in all structures. For eclipsed and inclined stacking motifs, it is simply the (0001) peak. For serrated and HCP-like staggered structures with two layers in the unit cell, the stacking peak is defined as (0002) (the (0001) peak vanishes, due to the intermittent-layer effect described before). For FCC-like staggered structures employing a unit cell like in Figure 2, that unit cell contains three layers, such that the stacking peak corresponds to the (0003) reflection. A second relevant aspect is that our simulations show that the interlayer distance *d_z_* is decreased by slips between the layers as a consequence of the significant bending of adjacent layers, as illustrated in Appendix A. This effect is particularly pronounced for staggered stacking, as there, by definition, nodes of one layer are positioned above the pores of neighboring layers. This enables layers to bend into the available space, reducing the overall stacking distance. For FCC-like staggered stacking, this results in a remarkably small stacking distance of *d_z_* = 2.70 Å (and a correspondingly large Bragg angle 2θ = 33.2°). In contrast, for eclipsed stacking, the layers are forced to be planar by their neighboring layers, resulting in a 40% increase in the stacking distance. As a consequence, the stacking peak (0001) is located at 2θ = 23.5° (*d_z_* = 3.78 Å), corresponding to an exceptionally large stacking distance compared to previous experimental (3.33 Å) as well as theoretical (3.35 Å) investigations [1,27]. This is not unexpected, considering steric arguments and the strong Pauli repulsion between eclipsed layers [27]. This large discrepancy between the predicted and measured peak positions, combined with the relatively high energy due to maximized Pauli repulsion, makes the eclipsed structure an unlikely candidate for the actual structure of LCOFs, in general, and COF-1 in particular.

The serrated stacking motif, however, has a stacking peak at 2θ = 23.5° (*d_z_* = 3.28 Å), as an ‘intermediate’ state with a somewhat reduced steric hindrance and Pauli repulsion resulting from the intermediate slip of ***s*** = (3.7, 1.0) Å. Thus the stacking distance of the serrated structure displays a near perfect agreement with the experiment. Also, the (10-11) peak of the serrated motif aligns well with the third, rather minor peak observed experimentally. Additional peaks can be regarded as too weak to be well resolved in the experimental pattern.

The main deviation that remains is the highly asymmetric shape of the {10-10} peak. This observation has already been attributed to disorder [24,28]. It suggests that the actual structure of COF-1 consists of slipped layers, but that slips between consecutive layers likely occur in different, potentially more or less random directions. Before developing an approach to build structural models including stacking disorder, it is useful to more systematically study how the distortion of the unit cell due to varying magnitudes and directions of interlayer slips impacts the PXRD results for LCOFs.

### 2.2. A PXRD Reference Guide for Slipped LCOFs

To gain a more comprehensive insight, we will, in the following, discuss how the PXRD patterns change upon varying the direction and magnitude of the slip. Here, the focus will be on inclined structures (and its derivatives; see above), as for these the slips have the most profound impact on PXRD spectra. To complement the discussion, also serrated structures will be considered. There, the unit cells (and, thus, the peak positions) are independent of the slip, while the structure factors and, thus, the intensities of the various peaks, change upon varying the layer alignment. In real systems, a certain inter-layer slip occurs depending on the type of LCOF, on linker side groups, and on pore adsorbates [29,30]. In silico, however, we can enforce any slip for any structure. In this way, we can generate a dataset that will make it more straightforward to infer a certain type of slip based on the PXRD data measured for a hitherto unknown crystal.

In this context, Figure 5 shows the evolution of the PXRD pattern for slips following certain paths. On the *y*-axis, we display the PXRD intensity plus the location along the path (either as a fraction of the linear path length, or of the angle along a radial path). The focus here is on a slip along a ‘horizontal’ path (0°; Figure 5a,b), for which at least certain inter-planar distances remain constant, and on two inclined paths (15° and 30°; Figure 5c–f), for which the distance between all crystallographic planes depends on the magnitude of the slip.

As expected, the graphs show large variations in the diffractograms, especially for the inclined structures. There, the degeneracy of the {10-10} peaks is lifted. For a slip angle of 0° (parallel to ***a***), the (01-10) peak remains unaffected by the slip, as the distance between the corresponding crystallographic planes does not change (see Figure 4b). In contrast, the same slip does reduce the distances between the (10-10) and (1-100) planes. For symmetry reasons, this happens to the same degree for both planes, as one can infer from Figure 4a,c. As a consequence, the corresponding peaks remain degenerate, which results in an overall splitting of the main peak into two peaks with a continuous shift of the (degenerate) (10-10) and (1-100) peaks to higher scattering angles for larger slips, i.e., the degree of splitting of the first two peaks is a measure for the magnitude of the horizontal slip.

In passing we note that the observation that the stacking peak ((0001) for inclined; (0002) for serrated) appears to be unaffected by slip is a consequence of the construction of the model system used for generating Figure 5: In contrast to the optimized structures used for generating Figure 3, for the more conceptual considerations in this section, the structures were built taking the atomic positions from the eclipsed structure without further optimization and setting the inter-layer distance to the experimental value of 3.3 Å.

For slips at angles deviating from 0°, the interplanar distances for all crystallographic planes associated with the first peak in the eclipsed configuration are reduced at least for small to intermediate slips. Consequently, all associated diffraction features shift to larger scattering angles (Figure 5c,e). If they still split up into only two peaks, this is an indication for a slip along a high-symmetry direction, as shown for a 30° slip in Figure 5e. As can be inferred from Figure 4, in that case the change in interplanar distance is the same for the (01-10) and the (1-100) planes, such that the corresponding peaks remain degenerate. Figure 4 also indicates that a diagonal slip should have the most pronounced impact on the (10-10) planes. This is the reason for the much stronger shift in the associated peak in Figure 5e. For slips in ‘less symmetric’ directions, no degeneracies prevail and, thus, three peaks appear as shown in Figure 5c. Notably, for any inclined slip direction one does not observe a mirror symmetry in the diffractograms for a relative slip of ½. The reason for that is that in those cases, the maximum slip vector (as indicated in the insets in Figure 5) is not a lattice vector.

The situation changes fundamentally for serrated structures. There, the unit cells (by definition) always have the exact same shape. Therefore, a change in slip has no impact on the interplanar distance and, thus, on the location of Bragg peaks. Still, also in serrated structures, the position of atoms within the unit cell change depending on the slip. The relative arrangement of atoms alters the structure factor, and, therefore, the intensity of the Bragg peaks. This can be clearly seen in the curves for serrated structures in Figure 5b,d,f. There, we see the three strong peaks caused by the {10-10} reflections (2θ ≈ 7°), by the {10-11} reflections (2θ ≈ 15°), and by the (0002) reflection (2θ ≈ 27°). The variation in relative intensities between those peaks is rather gradual and the actual intensity of the peaks in general also depends on the details of the chemical structure of an LCOF. This suggests that for serrated structures, the slip is not easily identified from PXRD data.

When following the evolution of the PXRD pattern upon varying slip directions between 0° and 60°, trends like that shown in Figure 5g are obtained (with the nature of the shifting peaks explained in Figure 5h). Considering such azimuthal paths is insofar relevant, as they often follow low energy valleys of the potential energy surface [27]. Figure 5g illustrates the situation for an inclined unit cell of COF-1, with a slip distance of 8.7 Å. This value has been chosen as the path then crosses the center of the hexagonal pore at 30°. As illustrated in Appendix A, changing the slip distance when varying slip directions mostly affects the range of scattering angles over which peaks shift, but it hardly affects the observed patterns. For serrated structures, one sees there that (at least for COF-1) not even the relative intensities of peaks are significantly impacted by changing the slip direction.

Regarding the trends in Figure 5g, one sees that for small angles, the (10-10) and (1-100) peaks are degenerate and appear at a scattering angle 2θ = 15°, while the (01-10) peak is found at 6.8°. Upon increasing the slip angle, the (01-10) peak shifts to larger scattering angles and the degeneracy of the (10-10) and (1-100) peaks is lifted, with the (1-100) peak shifting to smaller and the (10-10) peak shifting to larger scattering angles. As a consequence, for a slip direction of 30°, the (01-10) and the (1-100) peaks become degenerate, consistent with what has been discussed above for Figure 5e. For slips at larger angles, the trends are reversed such that the situation is again ‘symmetric’ with respect to a slip direction of 30°.

Besides providing guidelines for connecting PXRD patterns to stacking motifs of LCOFs, the data presented in this section indicate that disorder in inter-layer slips would typically result in increasing diffraction intensities at scattering angles larger than the 6.8° (corresponding to the {10-10} peaks of the eclipsed situation in COF-1). This implies that such disorder could well be responsible for the asymmetric shape of the corresponding peak in the experimental diffractogram of COF-1. To clarify what would be the most likely slips in a disordered structure, it is important to identify the conformations associated with (local) minimum structures and to determine their relative energies.

### 2.3. The Stacking Energy Landscape of COF-1

Thus, as a next step the stacking-energy landscape needs to be explored. This is done for COF-1 using quantum mechanical modeling to calculate the potential energy surface as a function of the inter-layer slip. These simulations have been performed combining 55 geometry optimizations with Gaussian process regression to interpolate between the training data [31,32]. A more in-depth description of the computational strategy is provided in the Methods section. The results are shown in Figure 6a,b for the inclined (blue) as well as for the serrated model structures (red). They reveal the existence of multiple local energy minima with energy barriers between them. The simulations also provide information on high-energy configurations. In passing we note that the energy as a function of stacking arrangement in LCOFs like COF-1 has been investigated before [20,27,33], but here we report a particularly accurately calculated PES with new local minimum structures. Most importantly, we use density functional theory with state-of-the-art van der Waals corrections (instead of more approximate methods), we employ converged basis sets and k-point grids and we fully relax our structures including atomic positions and lattice parameters. Following the procedure described in the Methods section, we obtain the total electronic energy, as well as the stacking distance *d_z_* (shown in Figure 6e,f) as a function of the slip (*s_x_*, *s_y_*). To better illustrate the local and global minimum configurations, we compiled so-called disconnectivity graphs using a flooding algorithm. In those graphs, displayed in Figure 6c,d, the open ends of vertical lines indicate the energy values (per unit cell and per LCOF layer) of local minima relative to the global minimum for both considered stacking motifs. The horizontal lines specify the heights of the lowest energy barriers connecting the minima. For the inclined configuration, the PES reveals that there are two (symmetry equivalent) slips (B and B’) that represent global energy minima. The disconnectivity graph shows that B and B’ are separated by a barrier of only 25 meV per unit cell. This suggests that barriers between minima could be thermally overcome at room temperature, possibly even post-synthesis (at least provided that the intra-layer correlation lengths are small enough—c.f., discussion in Section 2.4). The next lowest minimum (referred to as A) appears at 31 meV per unit cell above the global minimum, from which it is separated by a barrier of 52 meV. Minimum C is numerically detected, but can, for all practical purposes, be regarded as a saddle point, since it has a depth of less than 1 meV. Applying the six-fold symmetry of a COF-1 monolayer, one obtains 18 possible slipped configurations corresponding to the aforementioned minimum structures. This makes the idealized inclined stacking motif highly unlikely, and suggests that in actual COF-1 crystals, slips between adjacent layers will most likely occur in many different directions. In fact, the three slips A, B, and B’ all occur between 2 Å ≤ |*r*| ≤ 4 Å, where the energies per unit-cell are all within ~60 meV. This is also illustrated in Appendix A, which shows the energy as a function of slip angle at the slip distances of the A and B configurations.

At higher energies, we find three more minima. Slips D, D’, are close to and E corresponds to the staggered configuration. Their energies are about 65 meV above that of the minimum configuration and are separated from the group of lower minima by a barrier of 125 meV. As a consequence, these minima are barely occupied in a disordered COF-1 crystal. Furthermore, for the eclipsed structure the energy per unit cell and layer is by 426 meV higher than for the global minimum. As discussed already previously, this makes such a structure energetically clearly unfavorable [23,24,27]. This is primarily a consequence of Pauli repulsion [20,27]. In the context of inclined conformations, it is worthwhile mentioning that also none of the local minimum structures identified above results in a PXRD pattern that would match the experimental one, as explicitly shown in Appendix A.

The PES for the serrated configuration looks qualitatively similar, yet quantitatively different. For small displacements, the lowest minima, B and B’, are slightly higher in energy (by 8 meV) than their counterparts for the inclined structures. Serrated minimum A, however, is slightly lower in energy (22 meV) than its inclined equivalent. For larger slips, the differences are more pronounced: saddle point C appears at roughly the same radius, but at angles 30° higher and lower than for inclined structures. Similarly, the near-staggered minima D and D’ have moved to 0° and 60°. In contrast minimum E is still in the staggered position. As the probably most relevant difference, serrated minima C, D, and E have significantly larger energies compared to A and B than in the inclined structures. This raises the question, what causes these differences that imply that not only nearest-, but also next-nearest neighbor interactions can become relevant. At first glance, the latter might appear rather surprising especially in view of the fact that van der Waals forces as the most relevant source of attraction between COF layers are rather short ranged. A closer inspection of the geometries of the resulting COFs, however, reveals that the main cause for the difference are different evolutions of the stacking distances, as shown in Figure 6e,f and Appendix A: For the inclined configurations, the reduction in inter-layer distances is much more pronounced at large slips than for the serrated ones. The reason for that is the bending of the monolayers in the inclined structures, which results in larger van der Waals attractions. This observation is in line with what has been discussed in Section 2.1, namely that for the FCC-like staggered stacking (which is equivalent to minimum E for the inclined stacking mode) a much smaller average inter-layer distance of (2.70 Å) is observed than for the HCP-like staggered stacking (3.08 Å). A more in-depth discussion is provided in Appendix A.

### 2.4. Structural Models Including Disorder

The main conclusion of the previous section is that there exist many possible slips which are low in energy, and which are separated by small energy barriers. Thus, one expects a COF-1 crystal to have a high degree of disorder. This raises the question of whether the above data could be used to generate at least an approximate non-empirical model for such a disordered system. To that aim, we use the full energy landscapes of Figure 6a,b, and calculate an occupation probability for each displacement. This is done by applying a Boltzmann probability distribution considering the differences in energies of the various stacking conformations (for details see Method section). In that way, a probability density distribution for slips like the ones shown in Appendix A is obtained, where it is important to stress that for the actual structure determination the pattern shown there needs to be replicated to account for the six-fold rotational symmetry of the system.

The first question that arises in this context is which of the above energy landscapes to use as basis for the Boltzmann distribution. Neither the inclined nor the serrated stacking are expected to prevail in a realistic COF, especially considering that slips can occur in all possible directions with probabilities calculated from Figure 6a,b repeating every 60°. When taking the actual multilayer structure into account, one, however, faces a combinatorial explosion of possible arrangements, which renders reliable quantum-mechanical calculation of the relative energies (and, thus, likelihoods) futile. This essentially limits ab initio approaches to strategies building COF models in a layer by layer fashion [24]. Then, the probabilities for specific displacements of adjacent layers can be calculated on energy landscapes like the ones shown in Figure 6a,b. In this context, one benefits from the fact that the inclined and serrated conformations represent two limiting cases for maximized (inclined) and minimized (serrated) next-nearest neighbor interactions, such that the actual situation can be expected to lie within these limits.

Even more relevant is the question, for which building-block sizes the relative energies should be plugged into Boltzmann-weights. So far, energies per unit cell per layer have been considered without a clear formal justification for that choice. Notably, Zhang et al. proposed to use the energy of the molecular building unit of a COF to represent the “difference in energy cost of synthesizing the LCOF structures from the monomers” (see Supporting Information of [24]). In our case, this corresponds to choosing an energy scale that is given by 1/3 of the energy per layer per unit cell (i.e., choosing an energy scaling factor for the values shown in Figure 6a,b of *S_E_* = 1/3).

In contrast, when discussing the thermodynamics for the coexistence of different polymorph structures on surfaces, Heenen and Reuter suggest that a supercell should be used that exceeds the system’s correlation length [34]. Beyond that length-scale, the growth in energy upon increasing the size of the considered domain would be directly compensated by the fact that for independent sub-systems the individual grand-canonical partition functions become multiplicative such that the supercell size would no longer impact the probabilities [34]. To determine the correlation length corresponding to distances within the COF-layers beyond which the dynamics of sub-systems no longer influence each other, we performed molecular dynamics (MD) simulations [35] using machine-learned potentials produced by active learning within the Vienna Ab initio Simulation Package (VASP) [36,37,38]. Analyzing the respective trajectories (in particular the dynamic evolution of inter-layer slips) shows a clearly reduced correlation for 2 × 2 supercells and a converged behavior for 4 × 4 lateral supercells. Starting from this supercell size, the range of displacements in the MD runs becomes essentially independent of the supercell size. Thus, we also explored the situation encountered when rescaling the energies per layer per unit cell by factors of 4 and 16, respectively. For the sake of comparison (albeit in the absence of a firm physical justification), also an energy landscape rescaled to the energy per atom was considered as a lower boundary. This was achieved by multiplying the energy landscape by a factor of 1/42. The resulting normalized occupation distribution for all considered cases are visualized in histograms as a function of the slip directions and magnitudes in Appendix A. There, we see that for lower energies (e.g., energy landscapes scaled down by a factor of 1/3 or 1/42), an extensive variety of slip lengths and directions are observed (c.f., also Figure 7a). In contrast, for higher energies (e.g., scaling factors of 4 or 16) the probable slips are strongly concentrated around the global energy minima (see Figure 7b). For *S_E_* = 4 or *S_E_* = 16, this results in essentially twelve possible slips between adjacent layers (the minima B and B’ repeated every 60°).

To model diffraction data for a powder, and to get rid of statistical outliers, we generate 64 crystals, each consisting of periodic repetitions of 64 layers. The slips between the 64 layers followed the probability distributions based on the inclined and serrated energy maps and including the scaling factors described above. The averages of the simulated powder diffractograms are shown in Figure 8.

For all considered cases, instead of shifted, individual peaks like in Figure 5 and in Appendix A, one observes a smooth, but highly asymmetric main diffraction maximum in the range between 7° and 11°. This is consistent with the experimental data and confirms the notion that there is a significant degree of disorder in the stacking of the COF layers. For the smallest energy scaling factor (*S_E_* = 1/42), the asymmetry of the main peak is, however, clearly overestimated with significant scattering intensity also in the region between 10 and 20 Å. This is a consequence of an overestimation of long-distance slips due to the essentially flat energy landscapes, as indicated already in the histograms in Appendix A. For *S_E_* = 1/3, this effect is reduced but still present, which is also not unexpected: the energy per molecular building unit, is certainly a valid quantity to determine around which global minimum structures the slips are distributed during the growth process, but it yields a too flat energy landscape for describing thermally excited deviations from global minimum conformations. Starting from an energy scaling of 1, the shape of the main peak appears converged. This suggests that, as long as thermal deviations from global minima are not severely exaggerated (as in the aforementioned cases) their actual magnitude has only a rather small impact on PXRD data. This, however, also means that conventional PXRD data are not well suited for studying such thermal excitations.

As far as the stacking peak is concerned, when building on the inclined PES, the scattering angle for the stacking peak is overestimated especially for small energy scaling factors (see Figure 8a). This is not unexpected considering that inclined stacking results in small inter-layer distances, especially when slips are large (see Figure 6e). Correspondingly, for *S_E_* = 16, where particularly large slips are highly unlikely, the stacking peak is shifted to 2θ = 27.89°. This corresponds to an average interlayer distance of *d_z_* = 3.20 Å and represents only a rather small deviation from the experimental value of 3.28 Å.

In contrast, for crystals built using the serrated PES, the stacking distances are overestimated and decrease with increasing energy scaling (see Figure 8b). As a consequence, at the converged supercell size of 4 × 4 (i.e., for an energy-scaling factor of *S_E_* = 16), the simulated stacking peak lies at 27.14°, which perfectly matches the experimental one at 27.17° such that measured and simulated stacking distances amount to 3.28 Å.

Another aspect worth mentioning is that the weak experimental maximum at 15.1° is not clearly resolved in any of the considered disordered structures. According to Figure 3, it is, however, reproduced by the (10-11) peak of the fully optimized and periodic serrated structure. This suggests that the experimentally studied sample might contain a certain fraction of crystallites in which a serrated-style stacking prevails such that more or less opposite slip directions occur in consecutive layers. However, to further support such an assessment, additional experiments and simulations would be necessary.

A final question that needs to be addressed is to what extent the insights discussed above can be generally applicable to LCOFs. To answer this, it is useful to consider, which interactions determine the inter-molecular stacking. As discussed in detail in [27], for flat LCOFs these are primarily (i) the van der Waals attraction between adjacent layers, (ii) Pauli repulsion (maximized for eclipsed stacking), (iii) attractive charge-penetration effects, and (iv) classical electrostatic interactions [39]. When such a situation prevails, inclined and serrated stacking motifs with global minima for comparably small slips are to be expected. In [27], this has been shown for a variety of pore topologies, in particular also for COF-5 (as a more extended analog of COF-1), for two COFs consisting of square pores realized via porphyrin nodes [40], and for a COF comprising triangular pores with hexabenzocoronene nodes [41]. Also in [24], for several systems serrated stacking with small offsets (≈1.5–2 Å) has been identified as the favorable interlayer alignment. Due to symmetry reasons, slips between adjacent layers are then again expected to occur in different directions resulting in stacking disorder [24]., i.e., the above conclusions can be applied directly to those cases.

The situation becomes more involved, when considering, for example, the stacking of non-planar layers. There, steric interactions play a dominant role. As a prototypical example, Haldar et al. studied the stacking of a porous dithiine-linked covalent organic framework, where undulated layers are caused by a bending along the C−S−C bridge [42]. Notably, even in this case an inclined stacking motif was identified as the global minimum structure with various serrated stackings being energetically close. This is at least the case, when assuming a fully periodic structure. To assess the impact of disorder also for such and even more complex (e.g., interwoven) structures [43] the present manuscript at least outlines a promising strategy.

## 3. Methods

### 3.1. DFT Simulations

For all DFT calculations, we used FHI-aims version 240507 [44,45,46,47] with the Perdew-Burke-Ernzerhof functional [48], and the nonlocal many-body dispersion correction [49]. Spin was not considered, and a net total charge neutrality was enforced. The atomic ZORA approximation was used to account for relativistic effects [50]. For all atomic species the FHI-aims default ‘tight’ basis sets were applied. Unless otherwise specified, a 2 × 2 × 3 k-grid was used. These settings are the results of convergence tests aimed at ensuring a convergence of the total electronic energy to within 0.2 meV per atom (see Appendix A), where we expect energy differences between different conformations to be significantly more accurate. Self-consistency convergence settings for the total energy, charge density, eigenvalues, and forces were 10^−6^ eV, 10^−6^ e/a_0_^3^, 10^−3^ eV, 10^−4^ eV/Å, respectively. All geometry optimizations were performed using the trust radius enhanced BFGS optimization algorithm [51], as implemented in FHI-aims, with a force convergence threshold of 10^−2^ eV/Å. All DFT calculations were performed using the Austrian Scientific Computing (ASC) infrastructure. Computation files are available via the TU Graz repository (DOI: 10.3217/yvvbv-k2b26).

### 3.2. Generating Potential Energy Surfaces Using Gaussian Process Regression

First, the geometry of a COF-1 unit cell was optimized for an eclipsed configuration. This yields sheets of COF-1 without curvature, with all atoms lying in one plane. From that optimization, the lattice vectors were extracted and defined as ***a*** = (*a_1_*, 0, 0), ***b*** = (*b*_1_, *b*_2_, 0), and ***c*** = (0, 0, *d_z_*). For systematically varying the slip in the inclined conformation, the third lattice vector was subsequently set to ***c*** = (*s_x_*, *s_y_*, *d_z_*), with ***s*** = (*s_x_*, *s_y_*) defining the slip in *x* and *y* directions. Subsequently, also this structure was optimized while constraining ***a***, ***b***, *s_x_*, and *s_y_*, such that an energetic minimum with respect to the atomic positions and the stacking distance *d_z_* was obtained.

For serrated structures, a 1 × 1 × 2 supercell was generated with ***c*** set to (0, 0, 2*d_z_*). This yields a unit cell containing two layers, and alternating slips ***s*** and −***s***. The *x* and *y* coordinates of two equivalent atoms (one in each layer) are then constrained, so that the slip is preserved during the relaxation.

To obtain a dense grid on a large slip space (0 Å to 9.5 Å and 0° to 60°), we start with 25 randomly selected slips. Gaussian process regression (GPR) was used to interpolate between those training points by predicting the energy (and stacking distance). GPR has the advantage of being able to give a Bayesian error along with predicted values [31,32]. This feature can be exploited to efficiently select additional training points. Based on that, grid points with the highest uncertainties were continuously added, until 50 training points were reached. After that, to enhance the accuracy of the most relevant points, the slips corresponding to predicted local minima were added as additional training points. This process yields a PES with a maximum interpolation-derived uncertainty of 5 meV per layer and unit cell.

In this context, it needs to be mentioned that due to the complexity of the system, only differences in the electronic energy between the considered configurations were accounted for and contributions due to variations in the configurational entropy and in the vibrational enthalpy were neglected. However, it appears safe to assume that the variations in these quantities with slip would be significantly smaller than the variations in the electronic energy.

Since the exact geometric structure of slips other than the training points is unknown, we did not use geometry-derived descriptors for the GPR. Instead, *s_x_* and *s_y_* function as the descriptors, fed to a Radial Basis Function (RBF) kernel, as implemented in scikit-learn [52].

### 3.3. Idealized Geometries

The idealized geometries in Figure 3 were generated based on the same principle as the training points underlying the PES. For the eclipsed geometry, the slip was constrained to be ***s*** = (0, 0). For the inclined structure, the slip was constrained to be in the local minimum of the inclined PES; ***s*** = (3.7, 1.0) Å. For the serrated structure, the same procedure was followed with a slip of ***s*** = (3.6, 0.8) Å. As discussed in the previous subsection, a 1 × 1 × 2 supercell with ***c*** set to (0, 0, 2*d_z_*) ensures a serrated cell with alternating slips. For the HCP-like staggered structure, a serrated cell with a fixed slip of ***s*** = 2/3 ***a*** + 1/3 ***b*** was used. For the FCC-like staggered structure, an eclipsed cell with a slip of ***s*** = 2/3 ***a*** + 1/3 ***b*** was used. For the (partially constrained) geometry optimizations, a k-point grid of 3 × 3 × 7 points was used.

### 3.4. PXRD Reference Guide Geometries

The geometries underlying the PXRD patterns in the PXRD reference guide of Figure 4 were generated using the planar sheets of COF-1 obtained in the eclipsed geometry optimization described above. The desired slip was again induced by changing the lattice vector ***c*** to be ***c*** = (*s_x_*, *s_y_*, *d_z_*). Here, the experimental value of *d_z_* = 3.3 Å was chosen and was kept fixed. No geometry optimizations were performed for these geometries, as they would have had virtually no impact on the structure factors.

### 3.5. Boltzmann Distribution

The energies *E_i_* from the PES are representative for slips that are sorted into “pixels”, *i*. The pixels follow a polar distribution, with a linear spacing in both the slip distance *r* (with Δ*r* = 0.01 Å) and the slip angle φ (with Δφ = 0.5°). Each pixel is assigned a probability density *ρ_i_* according to a Boltzmann distribution (Equation (1)).(1)$$\rho_{i}=\frac{\exp\left(-\frac{E_{i}}{k_{B}T}\right)}{\sum_{i}A_{i}\exp\left(-\frac{E_{i}}{k_{B}T}\right)}$$

The densities are normalized, such that an integration of *ρ_i_* multiplied by the pixel size A*_i_* over all pixels contained in the lateral unit cell yields one. Concerning the values of *E_i_*, various sizes of the “energetically relevant” repeat unit of the COF were considered, as discussed above. In this context, it needs to be stressed that only differences in electronic energies were taken into account, as variations in vibrational energies (and entropies) of differently slipped structures are assumed to be of only minor relevance.

### 3.6. Disordered Geometries

The disordered geometries were constructed using planar sheets of COF-1. This is reasonable as (i) the impact of layer distortions on the potential-energy landscapes and stacking distances for the supercells would be intractable using ab initio methods, and (ii) the detailed conformation of the individual layers has no impact on the positions and hardly any impact on the intensities of the PXRD peaks. As a final step, the finite-size stacks are treated as supercells, employing periodic boundary conditions to eventually calculate diffraction patterns.

The PES for either inclined or serrated structures was used to calculate probability weights for slips with the considered energies representing fractions or multiples of the values obtained for unit cells, as described in results section. In this way, probability weights were assigned to all slips in the slip space. Consecutive layers are chosen according to this probability weight, and a randomly chosen rotation according to the six-fold rotational symmetry of the system was applied. This procedure is reminiscent of that pursued by Zhang et al., with a few differences: here, the full range of possible slips is considered, different energy scales are tested and the energy landscape has been calculated much more accurately (albeit only for a single COF rather than for a broad range of materials). The unit cell is determined from the succession of slips in the stacks, such that the system remains periodic in all three directions. This effectively generates a 1 × 1 × 64 supercell. The corresponding stacking distances *d_z_* between neighboring layers were chosen according to the optimized values reported in Figure 6e,f. This is in contrast to previous models, where a fixed theoretical or experimental stacking distance was chosen [20].

64 such crystals containing 64 layers were generated for each considered case, and their PXRD curves were averaged and then normalized.

### 3.7. PXRD Simulations

The powder X-ray diffractograms were calculated using an in-house written code. The intensity represents the square of the structure factor, corrected with the polarization factor for unpolarized X-rays, as well as the Lorentz factor [53]. The intensity as a function of the scattering angle 2θ was obtained by summing Gaussian functions with a fixed width (σ = 0.2°) over all Bragg reflections. This Gaussian broadening is meant to account for inherent peak broadening effects like finite size crystal effects.

## 4. Conclusions

Our study shows that COF-1, as a prototypical layered COF with a honeycomb structure, exhibits a highly disordered stacking arrangement rather than any of the idealized periodic motifs often assumed for such materials. By systematically comparing simulated PXRD patterns of idealized eclipsed, inclined, serrated, and staggered stackings to experimental data, we rule out all high-symmetry periodic configurations as the dominant structural motifs for COF-1.

To bridge the gap between theory and experiment, as a next step we generated a comprehensive ‘PXRD reference guide’ for slipped LCOF structures, mapping the evolution of diffraction features onto both the, direction and magnitude of layer displacements. This links specific peak positions, peak splittings, and intensity changes to distinct stacking modes, enabling a more informed interpretation of experimental patterns for a wide range of LCOFs.

Potential energy surfaces for both inclined and serrated configurations calculated using ab initio techniques and augmented by Gaussian Progress-Regression reveal a landscape of numerous local minima separated by small energy barriers. Many of these minimum structures (corresponding to different slip magnitudes and directions) could be energetically accessible at room temperature and would also be realized during materials synthesis. This suggests that a significant stacking disorder is inevitable.

Based on the computed energy landscapes, we developed static, non-empirical models for disordered COF structures, in which slip directions and stacking distances are populated according to Boltzmann-weighted probabilities considering differently rescaled energy landscapes. PXRD patterns simulated from these models excellently reproduce not only the position, but especially the asymmetry of the main peak in the experimental PXRD data of COF-1 without any external input testifying to the potential of the ab initio-based simulations.

The workflow presented here provides a foundation for more accurate property predictions of LCOFs and offers a transferable strategy for other layered materials where stacking disorder plays a key role. Future extensions should incorporate finite-temperature dynamics to describe not only the static distribution of stacking modes, but also their time dependent evolution.

## Figures and Tables

**Figure 1 ijms-26-09222-f001:**
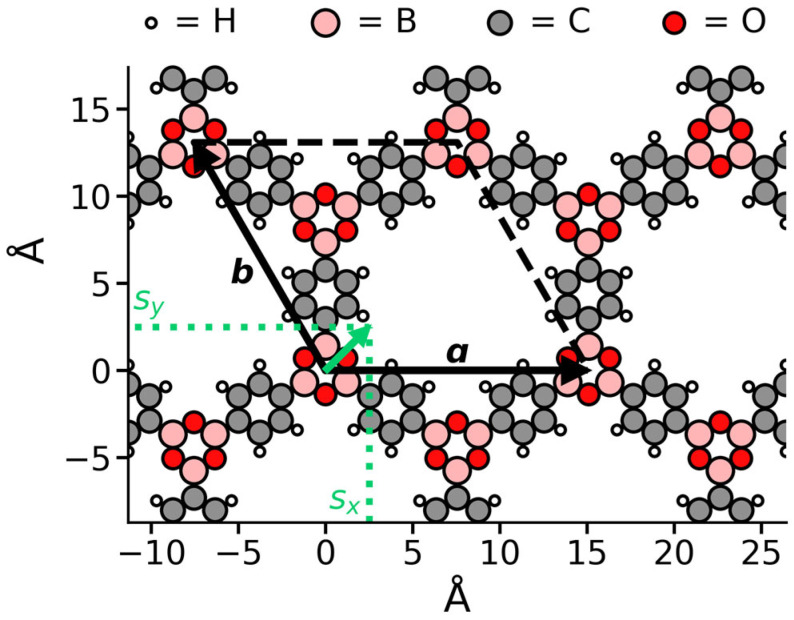
Idealized eclipsed crystal structure of COF-1. The lattice vectors ***a*** and ***b*** are denoted by black arrows, the dashed black line completes the 2D unit cell boundaries. The definition of the slip is indicated by the green arrow and its coordinates *s_x_* and *s_y_* are illustrated by the green dotted line.

**Figure 2 ijms-26-09222-f002:**
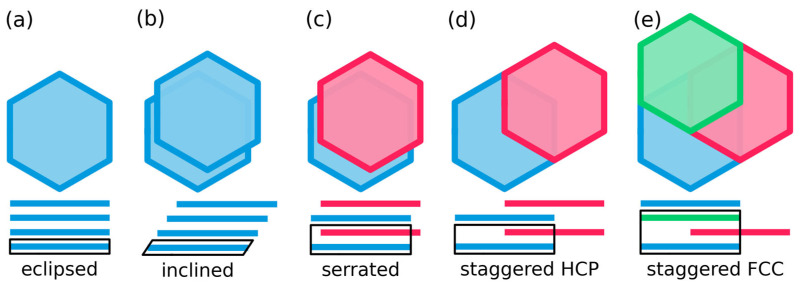
Top view and side view of idealized stacking modes for four layers of COFs. The stacking is illustrated by the in the present case hexagonal building blocks of the layers. Different colors illustrate different layers within one unit cell. Repeating colors illustrate periodic replicas in neighboring unit cells. In the side view, the 2D unit cell spanned by vectors a and c is drawn in black. (**a**) eclipsed stacking; (**b**) inclined stacking; (**c**) serrated stacking; (**d**) HCP-like staggered stacking (ABABAB); and (**e**) FCC-like staggered stacking (ABCABC).

**Figure 3 ijms-26-09222-f003:**
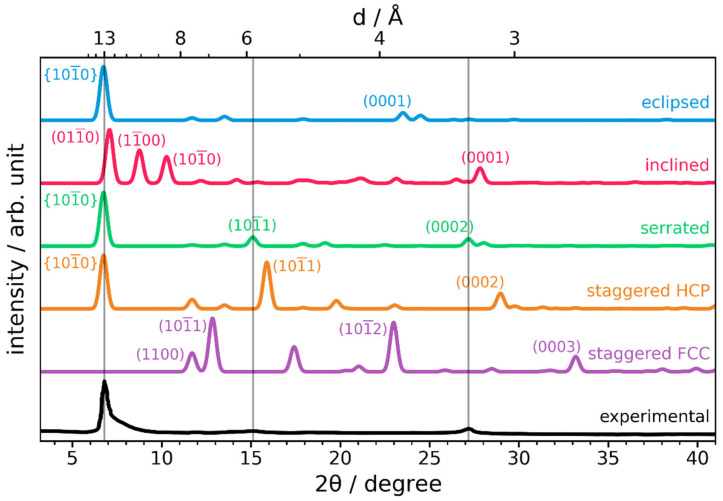
Normalized simulated powder X-ray diffractograms of idealized structures: eclipsed (blue), inclined (red), serrated (green), HCP-like staggered (orange), and FCC-like staggered (purple). The experimental diffractogram is plotted in black [1]. Vertical gray lines indicate important peak positions observed in experiment.

**Figure 4 ijms-26-09222-f004:**
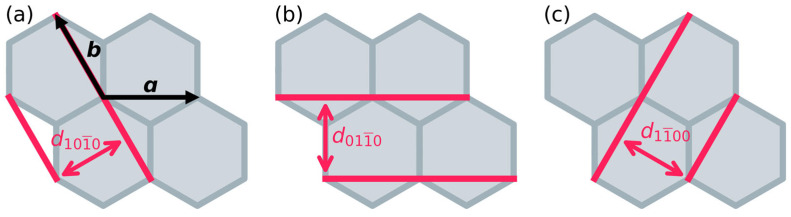
Illustration of the reflection planes and the corresponding distances of the (10-10) (**a**), (01-10) (**b**), and (1-100) (**c**) reflections indicated in red. The lattice vectors ***a*** and ***b*** are indicated by black arrows.

**Figure 5 ijms-26-09222-f005:**
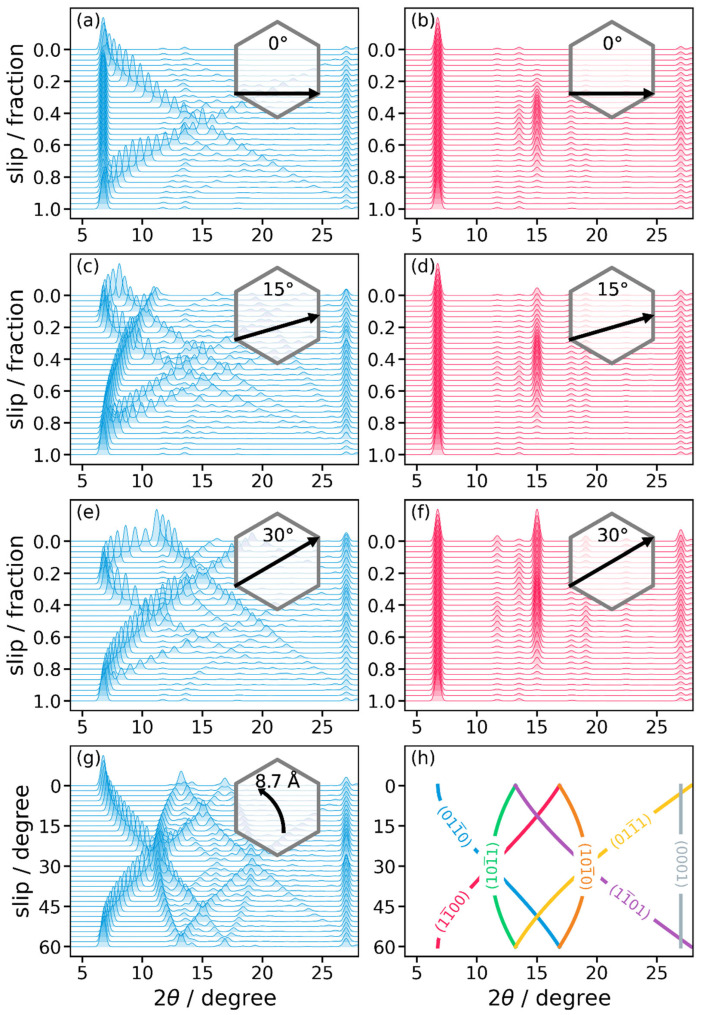
Evolution of X-ray powder diffractograms as a function of the slip along pre-defined paths through the unit cell. These paths are marked by a black arrow in the insets. Inclined structures are plotted in blue (**a**,**c**,**e**,**g**), serrated structures in red (**b**,**d**,**f**). The lengths of the black arrows also define the slip distances that correspond to a ‘relative fraction’ of one. For slips along ***a***, for symmetry reasons the diffraction pattern is mirror symmetric with respect to a relative slip of 0.5; this is not the case for the other displayed slip directions. For panel (**g**), the direction of the slip is varied, with the magnitude of the slip fixed at to 8.7 Å. Panel (**h**) serves to illustrate the nature of the peaks that dominate the diffractogram in panel (**g**). All results have been obtained for a COF-1 model with the original atomic positions determined for an eclipsed structure and with the stacking distance constrained to the experimental value of 3.3 Å.

**Figure 6 ijms-26-09222-f006:**
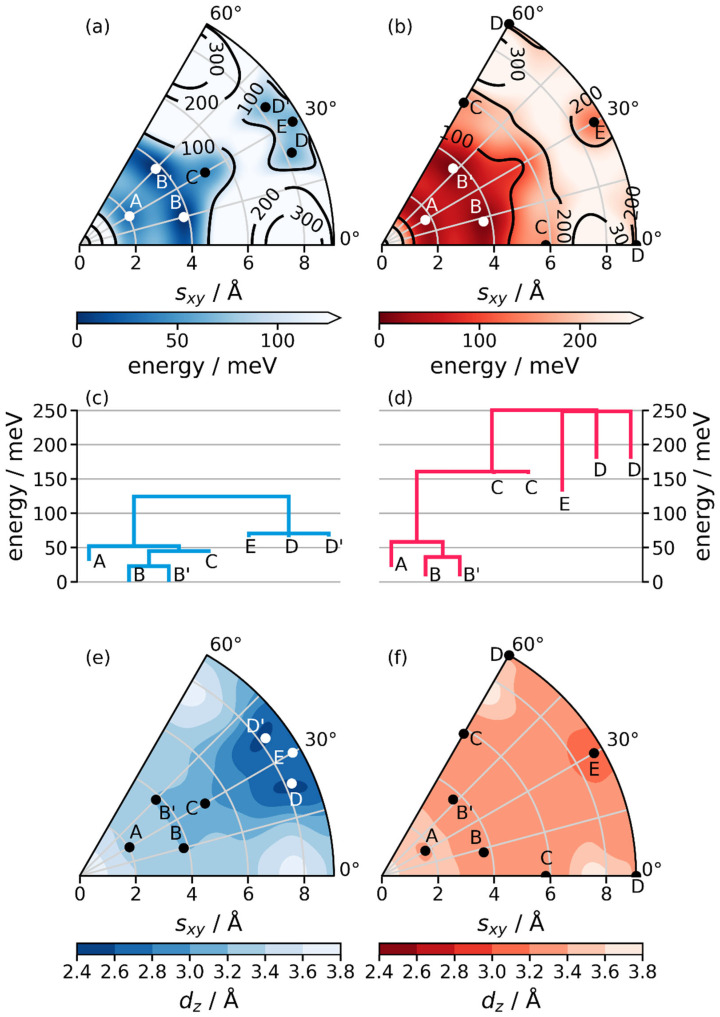
(**a**,**b**): full energy landscape within a 60° wedge, which is periodically repeated due to the hexagonal symmetry of the building blocks of the COF; the displayed quantity is the total electronic energy per unit cell per layer as a function of slip between two adjacent layers for the inclined (**a**) and serrated (**b**) conformations. Local minima are indicated by white or black dots and labeled with capital letters. Equal letters denote equivalent minima throughout the panels. (**c**,**d**): disconnectivity graph displaying the energy of the local minima in the PES (lower end of vertical lines) and the barriers between them (horizontal lines). (**e**,**f**): stacking distance *d_z_* as a function of slip for inclined (**e**) and serrated (**f**) conformations.

**Figure 7 ijms-26-09222-f007:**
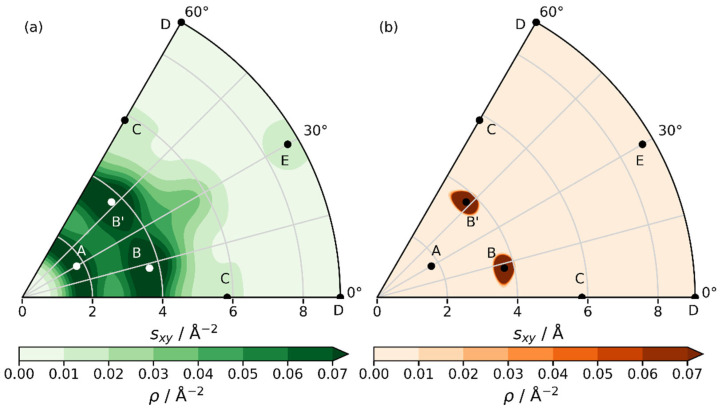
Probability density for the occurrence of specific slips as a function of slip magnitude and direction for energy scaling factors *S_E_* = 1/3 (**a**) and *S_E_* = 16 (**b**). The higher the energy scaling factor, the more concentrated the probability density becomes around the global minima.

**Figure 8 ijms-26-09222-f008:**
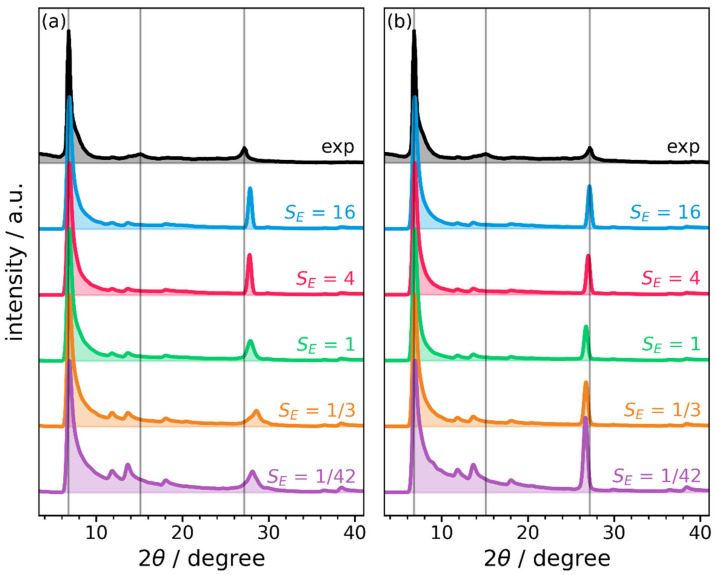
Simulated PXRD patterns of supercells containing 64 layers. Each displayed diffractogram has been obtained by averaging over 64 calculations for independently generated supercells. Different energy scaling factors *S_E_* were used to set the energy landscape for thermally populating all possible conformations through a Boltzmann distribution (for details see main text). Panel (**a**) shows the results for energies from a PES calculated for inclined structures and (**b**) for a PES calculated for serrated structures. The experimental PXRD curve is plotted in black, and the vertical lines indicate the locations of the most important experimental peaks.

## Data Availability

The following DFT computation files are available via the TU Graz repository at https://repository.tugraz.at/records/yvvbv-k2b26 (accessed date: 14 August 2025; DOI: 10.3217/yvvbv-k2b26).

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
