# Peer review of "Stacking in Layered Covalent Organic Frameworks: A Computational Approach and PXRD Reference Guide"

_ijms, 2025, doi:10.3390/ijms26189222_

Round 1
Reviewer 1 Report
Comments and Suggestions for Authors
The paper by Steentjes and Zojer presents an interesting workflow to characterize stacking in layered COFs, using COF-1 as a prototype. DFT with additional vdW treatment is combined with Gaussian-process interpolation to generate a stacking energy landscapes. This reveals multiple symmetry-related minima, which are separated by small barriers. A number of energy scans with varying slip directions provide already some hints on the possible asymmetry of the experimental PXRD peak at low angle. The low energy barriers further indicate possible thermal stacking disorder. Boltzmann-weighted sampling over these landscapes is finally used to construct disordered multilayer models from which PXRD patterns are computed. The resulting diffractograms closely match experiment, capturing peak positions, stacking distances, and the hallmark asymmetric main peak. Overall, the study provides a computational framework that clarifies the inherently disordered stacking of 2D-COFs and improves structural interpretation beyond idealized assumptions that are typically necessary in current literature. As such it can act as a blueprint for future studies.
The novelty lies in the rigorous and comprehensive analysis of the potential energy surfaces of the COF material. While stacking disorder in COFs is widely acknowledged, it is common practice in the literature to consider only staggered, inclined, or eclipsed stacking geometries when developing structural models for subsequent electronic structure calculations. This work provides guidance on how to better approach this initial step in theoretical modelling, which is what I mean by “blueprint” above. Additionally, the methods incorporate novel approaches such as Gaussian-process-based techniques to reduce the computational burden of numerous DFT calculations, which is also inspiring. Its presentation is very didactic, and I would certainly recommend it to students performing PXRD simulations for 2D-COFs once published. It will help them better understand the relationship between reflection positions and structure, as well as how to identify improved models.
The only comment, I have to make concerns the axis label of Fig. 5, which was a bit confusing, particularly the “+” symbol. As an optional suggestion, I believe one can remove int/a.u. and change the caption from PXRD data to “spectra”. This should be clear to the reader.
For these reasons, I recommend publication of the manuscript.
Author Response
We thank the reviewer for the extremely positive assessment of our manuscript.
Comment 1: The only comment, I have to make concerns the axis label of Fig. 5, which was a bit confusing, particularly the “+” symbol. As an optional suggestion, I believe one can remove int/a.u. and change the caption from PXRD data to “spectra”. This should be clear to the reader.
Response 1: We have changed the figure as suggested by the reviewer. In the caption we replaced “PXRD data” with “X-ray powder diffractograms”. We used this term instead of the term spectra suggested by the reviewer, as our colleagues working on x-ray diffraction discourage the use of the term spectra for diffraction experiments.
Reviewer 2 Report
Comments and Suggestions for Authors
Covalent organic frameworks (COFs) are currently highly favoured by researchers. Among them, the interlayer stacking structure of two-dimensional covalent organic frameworks has been controversial. This manuscript entitled "Stacking in layered covalent organic frameworks: a computational approach and PXRD reference guide" by Zojer et al. systematically investigates the interlayer stacking disorder in two-dimensional layered COFs using COF-1 as the model system, combining first-principles calculations with powder X-ray diffraction (PXRD) simulations, and proposes a novel structural modelling method that does not require empirical parameters. This work focuses on theoretical calculations and methodology, providing significant reference for future structural determination and performance studies of two-dimensional layered COF materials. However, the manuscript has the following issues:
1. The manuscript only studied one sample, COF-1, making it difficult to demonstrate the universality of this method.
2. The interlayer stacking structure of grid-like layered COFs is more complex and urgently needs to be determined.
3. It is strongly recommended to use ultra-high resolution synchrotron radiation diffraction data for comparison and discussion to avoid missing and overlapping diffraction signals.
Author Response
We thank the reviewer for the very positive assessment of our paper.
Comments 1 and 2:
However, the manuscript has the following issues:
1. The manuscript only studied one sample, COF-1, making it difficult to demonstrate the universality of this method.
2. The interlayer stacking structure of grid-like layered COFs is more complex and urgently needs to be determined.
Response 1 and 2: At the end of the results and discussion section we now explain in detail that for certain types of LCOFs one can expect a fully equivalent situation as in COF-1 (studied here). Regarding the mention of grid-like layered MOFs, we are not exactly sure, what the reviewer is referring to. Therefore, we now briefly compare the situation for planar COF layers to that of undulated layers and interwoven COFs. The corresponding section reads:
“A final question that needs to be addressed is to what extent the insights discussed above can be generally applicable to LCOFs. To answer this, it is useful to consider, which interactions determine the inter-molecular stacking. As discussed in detail in [27], for flat LCOFs these are primarily (i) the van der Waals attraction between adjacent layers, (ii) Pauli repulsion (maximized for eclipsed stacking), (iii) attractive charge-penetration effects, and (iv) classical electrostatic interactions [39]. When this applies, serrated stacking motifs with global minima for comparably small slips are to be expected. In [27], this has been shown for a variety of pore topologies, in particular also for COF-5 (as a more extended analogue of COF-1), for two COFs consisting of square pores realized via porphyrin nodes [40], and for a COF comprising triangular pores with hexabenzocoronene nodes [41]. Also in [24], for several systems serrated stacking with small offsets (≈ 1.5 - 2 Å) has been identified as the favorable interlayer alignment. Due to symmetry reasons, slips between adjacent layers are again expected to occur in different directions resulting in stacking disorder [24]. I.e., the above conclusions can be applied directly to those cases.
The situation becomes more involved, when considering, for example, the stacking of non-planar layers. There, steric interactions play a dominant role. As a prototypical example, Haldar et al. studied the stacking of a porous dithiine-linked covalent organic framework, where undulated layers are caused by a bending along the C−S−C bridge [42]. Notably, even in this case an inclined stacking motif was identified as the global minimum structure with various serrated stackings being energetically close. This is at least the case, when assuming a fully periodic structure. To assess the impact of disorder also for such and more complex (e.g., interwoven) structures [43] the present manuscript outlines a promising strategy.”
Comment: 3. It is strongly recommended to use ultra-high resolution synchrotron radiation diffraction data for comparison and discussion to avoid missing and overlapping diffraction signals.
Response 3: We agree to the reviewer that higher quality experimental data would be desirable. Unfortunately, we are not aware of any ultra-high resolution synchrotron radiation diffraction data for COF-1, which as the prototypical COF material is in the focus of our research.
Nevertheless, we adapted our manuscript providing a more in-depth assessment of how higher-quality insights could be generated using x-ray diffraction and we also mention that ultra-high resolution synchrotron radiation diffraction experiments could improve the situation. The added section reads:
“To experimentally analyze LCOF structures, one would ideally apply single-crystal diffraction experiments to sufficiently large COF crystals. These are, however hard to obtain and, thus, only a few years ago the first such study has been reported [17]. As an alternative approach, when studying thin films rather than large crystals, rotating grazing incidence diffraction performed at synchrotrons can be a viable tool for generating a picture of large fractions of reciprocal space of porous materials [18] and the technique could also be applied to COF thin films. Unfortunately, however, COFs are usually synthesized as powders comprising small crystallites [19]. Moreover, due to the aforementioned stacking disorder, single crystals with 3D periodicity are an unlikely scenario for many layered COFs. Thus, powder-diffraction (PXRD) experiments are still the most commonly applied technique and also in the current manuscript we will rely on comparing our simulations to that type of experimental data [20,21]. In passing we note that ultra-high-resolution synchrotron radiation diffraction could be advantageous to avoid missing and overlapping diffraction signals.”